# Synergistic Effects of Sanglifehrin-Based Cyclophilin Inhibitor NV651 with Cisplatin in Hepatocellular Carcinoma

**DOI:** 10.3390/cancers14194553

**Published:** 2022-09-20

**Authors:** Sonia Simón Serrano, Michele Tavecchio, Josef Mallik, Alvar Grönberg, Eskil Elmér, Chamseddine Kifagi, Philippe Gallay, Magnus Joakim Hansson, Ramin Massoumi

**Affiliations:** 1Translational Cancer Research, Department of Laboratory Medicine, Lund University, Medicon Village, SE-223 63 Lund, Sweden; 2Abliva AB, Medicon Village, Scheelevägen 2, SE-233 81 Lund, Sweden; 3Mitochondrial Medicine, Department of Clinical Sciences, Lund University, BMC A13, SE-221 84 Lund, Sweden; 4NGS & OMICS Data Analysis (NODA) Consulting, Flöjtvägen 10b, SE-224 68 Lund, Sweden; 5Department of Immunology & Microbial Science, The Scripps Research Institute, La Jolla, CA 92037, USA

**Keywords:** synergy, cyclophilin, PPIase, apoptosis, hepatocellular carcinoma, cisplatin

## Abstract

**Simple Summary:**

Cyclophilins, one of the three families of proteins with peptidyl-prolyl isomerase activity, are overexpressed in several cancers, including hepatocellular carcinoma (HCC), and this overexpression is correlated with poor prognosis. Cyclophilins play an important role in cancer progression; this role includes mediating chemoresistance. In this study, the effect of cyclophilin inhibition in HCC cells was evaluated to find potential combination treatments. We demonstrated that the novel cyclophilin inhibitor, NV651, reduced cell division and DNA repair. In addition, the combination of NV651 and cisplatin, a DNA damage reagent, can be considered an interesting novel treatment therapy for HCC as it significantly increases cancer cell death compared with that caused by cisplatin alone.

**Abstract:**

Hepatocellular carcinoma (HCC), commonly diagnosed at an advanced stage, is the most common primary liver cancer. Owing to a lack of effective HCC treatments and the commonly acquired chemoresistance, novel therapies need to be investigated. Cyclophilins—intracellular proteins with peptidyl-prolyl isomerase activity—have been shown to play a key role in therapy resistance and cell proliferation. Here, we aimed to evaluate changes in the gene expression of HCC cells caused by cyclophilin inhibition in order to explore suitable combination treatment approaches, including the use of chemoagents, such as cisplatin. Our results show that the novel cyclophilin inhibitor NV651 decreases the expression of genes involved in several pathways related to the cancer cell cycle and DNA repair. We evaluated the potential synergistic effect of NV651 in combination with other treatments used against HCC in cisplatin-sensitive cells. NV651 showed a synergistic effect in inhibiting cell proliferation, with a significant increase in intrinsic apoptosis in combination with the DNA crosslinking agent cisplatin. This combination also affected cell cycle progression and reduced the capacity of the cell to repair DNA in comparison with a single treatment with cisplatin. Based on these results, we believe that the combination of cisplatin and NV651 may provide a novel approach to HCC treatment.

## 1. Introduction

Liver cancer is classified as the sixth-most-common type of cancer, with an incidence of 905,677 new cases, and is the third-most-common cancer-related cause of death, with a toll of 830,180 deaths in 2020 [1]. Viral infections, such as hepatitis B or C virus, aflatoxin exposure, chronic alcohol consumption or non-alcoholic fatty liver disease, can lead to the appearance of hepatocellular carcinoma (HCC), the most common type of primary liver cancer [2]. The Barcelona Clinic Liver Cancer (BCLC) system facilitates the diagnosis and treatment of HCC patients [3]. According to the BCLC, patients diagnosed at a very early or early stage (stages 0 and A, respectively) can receive curative treatments, such as a resection, transplantation, or radiofrequency ablation (RFA), followed by transarterial chemoembolization (TACE), treatment with sorafenib (a tyrosine kinase inhibitor (for cancer stages B–C)) and symptomatic treatment (for stage D cancer) [4]. TACE has a dual function of occluding the hepatic arteries, blocking the blood flow and inducing tumour necrosis, as well as accumulating chemo agents in the tumour area [5]. During TACE, chemo agents, such as mitomycin, doxorubicin and cisplatin are widely used [3]. Cisplatin is a DNA damage reagent targeting both mitochondrial and nuclear DNA. It causes interstrand or intrastrand crosslinking [6] and has the capacity to increase the production of reactive oxygen species (ROS) [7]. Unfortunately, tumours can often acquire resistance to chemotherapy agents, such as cisplatin [8].

In recent years, other treatments, including lenvatinib [9], have been accepted for first-line systemic therapy, while for second-line systemic therapy, regorafenib [10], cabozantinib [11] and ramucirumab [12] have been approved. Unfortunately, these novel compounds can only prolong life expectancy by a few months before resistance develops [9,10,11,12,13,14], indicating that low objective response rates demand further improvement, which might be achieved by exploiting potential synergistic strategies. Combination therapies between immune checkpoint inhibitors (ICIs) increasing the T cell response and other anticancer agents have recently emerged as a new treatment approach [15]. In patients with unresectable HCC, atezolizumab, an ICI targeting PD-L1, and the antiangiogenic monoclonal antibody targeting VEGF, bevacizumab, have been shown to increase overall survival in comparison with a sorafenib-treated patient group [16]. These results show the potential of combinatory treatments against HCC and the importance of finding new targets for the treatment of HCC.

Cyclophilins, one of three families of PPIase proteins (together with parvulins and FKBPs) [17], are intracellular proteins with the capacity to catalyse the cis–trans isomerization of peptide bonds in proline residues [18]. Cyclophilins are found in all types of cells and organisms [19]. Multiple cyclophilins with different functions and intracellular locations have been identified in the human genome [19,20]. The most studied of them, cyclophilin A, B and D, reside mainly in the cytosol, endoplasmic reticulum and mitochondria, respectively [21,22,23]. Cyclophilins are involved in several functions, including protein-folding, and can act as molecular switches by activating or deactivating targeted proteins [19]. Cyclosporin A (CsA), the first natural cyclophilin inhibitor discovered, is currently used in clinical settings due to its immunosuppressant activity that works through the formation of a ternary complex with cyclophilin A and calcineurin, decreasing the activity and proliferation of T-lymphocytes [24,25]. Sanglifehrin A (SfA) is another natural cyclophilin inhibitor, and it has 20-fold higher affinity than CsA [26]. Despite the fact that cyclophilins are found in all cells [19], their overexpression has been observed in several types of cancer, including HCC, and they play a key role in several activities, including proliferation and cell cycle regulation [27,28,29,30]. In addition, cyclophilins have been shown to increase the expression of ABC transporters, thus decreasing intracellular drug accumulation and potentiating therapy resistance [31]. Due to their capacity to protect cancer cells against an increment in the ROS levels, cyclophilins have also been identified as potent antioxidants [32].

Due to their potential role in chemoresistance, several publications evaluating the effect of cyclophilin inhibition in combination with chemo agents, such as cisplatin, have emerged. This has resulted in a synergistic effect that increases cell death in HCC and ovarian cisplatin-resistant cancer cells, which is probably due to the inhibition of the antioxidant activity and potential decreased expression of genes involved in DNA damage repair [33,34].

In a previous study, we presented NV651, a new cyclophilin inhibitor based on the sanglifehrin scaffold. By performing a detailed characterization of the NV651 drug class, we concluded minimal inhibition of classical CsA off-targets in comparison with SfA [35]. We demonstrated that NV651 was a more potent cyclophilin inhibitor than CsA or SfA for inhibiting the PPIase activity of several cyclophilins; additionally, it exhibited a higher anti-proliferative capacity than sorafenib and displayed a capacity to decrease tumour growth in vivo. We observed an accumulation of cells in the G_2_/M specifically in the mitotic phase, which could potentially explain this potent antiproliferative activity. In addition, the safety of NV651 in normal cells and its good oral bioavailability were confirmed [36].

In the present study, we aimed to further investigate NV651’s mechanism of action via transcriptome analysis to understand NV651′s efficacy against HCC. We can confirm a decrease in the pathways involved in mitosis, which agrees with our previous results [36], and a decrease in several pathways involved in DNA damage repair. Due to this effect, we decided to evaluate the potential synergy with other treatments that are already used against HCC. The synergistic effect of NV651 and cisplatin on increasing cell death demonstrated this co-treatment as a potential combination treatment against HCC.

## 2. Materials and Methods

### 2.1. Cell Lines and Drugs

HEPG2 cells were purchased from the American Type Culture Collection (ATCC) (Manassas, VA, USA). HUH7 cells were obtained from the Japanese Collection of Research Biosources Cell Bank (Osaka, Japan). The cell lines used for the analysis of sensitivity markers were licensed from ATCC by Crown Bioscience (Suzhou, China). HEPG2 cells were authenticated by Short Tandem Repeat (STR) profiling. Experiments were performed between passages 2 or 3 up to passage 20 after thawing. HEPG2 cells were maintained in a high-glucose DMEM medium (Thermo Fisher Scientific, Cat# 11965-092, Waltham, MA, USA) supplemented with 10% foetal bovine serum (FBS) (Sigma Aldrich, Cat# F7524, St. Louis, MO, USA) and 1% penicillin/streptomycin (Thermo Fisher Scientific, Cat# 10378-016, Waltham, MA, USA). HUH7 cells were maintained in a low-glucose DMEM medium (Corning, Cat# 10-014-CMR, Corning, NY, USA) supplemented with 10% FBS and 1% penicillin/streptomycin. The cells were stored in a humified incubator at 37 °C under a 5% CO_2_ atmosphere. Trypsinization was performed with 0.25% trypsin (Corning, Cat# 25-053-CI, Corning, NY, USA). All cell lines were tested for Mycoplasma with the MycoAlertTM Mycoplasma Detection Kit (Lonza, Cat# LT07-418, Basel, Switzerland). Cisplatin (Sigma Aldrich, Cat# C2210000, St. Louis, MO, USA) was dissolved in 0.9% NaCl in MilliQ and stored at RT for up to 2 weeks in the dark. NV651 was dissolved in DMSO at 0.5 or 10 mM and stored at −20 °C until use. Sorafenib (Selleck Chemicals, Cat# BAY 43-9006, Houston, TX, USA) and Doxorubicin (Sigma Aldrich, Cat# D1515-10MG, St. Louis, MO, USA) were diluted in DMSO and stored at −20 °C. Mitomycin (Sigma-Aldrich, Cat# 10107409001, St. Louis, MO, USA) was dissolved in MilliQ-water and kept at −196 °C for long-term storage and +4 °C for short-term storage.

### 2.2. Analysis of Sensitivity Biomarkers

#### 2.2.1. End-Point Calculation and Comparison

The 50 cell lines comprised 3 tumour types: 10 colorectal cancer, 31 liver cancer and 9 pancreatic cancer. Cells were seeded with their respective culture medium in two 96-well plates with a final cell density of 4 × 10^3^ cells/well and left to attach overnight. The next day, T0 was analysed by the CellTiter Glo Luminescent Cell Viability Assay (Promega, Madison, WI, USA), performed according to the manufacturer’s instructions using an EnVision Multi-Label Reader. The second plate was treated with NV651 with 3.16-fold serial dilutions and analysed by CellTiter Glo after 72 h of treatment.

Dose–response curves were fitted by the 4-parameter model
(1)$$Inhibition\%=\mathrm{Bottom}+\frac{\mathrm{Top}-\;\mathrm{Bottom}}{1+10^{\left({\mathrm{logEC}}_{50}-x\right)\times \;\mathrm{HillSlope}}}$$
where Top and Bottom are the two asymptotes of the sigmoidal curve, EC50 is the relative IC50 and concentration *x* is in log-10 scale. To accommodate experimental errors, Bottom was allowed to go down to −20% and Top go up to 120%. The fitting error of a model was measured by $\frac{s_{EC50}}{EC50}$, where $\sigma_{EC50}\;$ is the standard error of EC50. In general, such a fitting error should be less than 40% for a model to be considered acceptable. The fitted area under curve (AUC) was calculated by
(2)$$AUC=\int_{a}^{b}\left(\mathrm{Bottom}+\frac{\mathrm{Top}-\;\mathrm{Bottom}}{1+10^{\left({\mathrm{logEC}}_{50}-x\right)\times \;\mathrm{HillSlope}}}\right)dx$$
where *a =* log (1, 10) and *b* = log (10,000, 10).

In the 50 cell lines, IC50 values were not obtained for 18 because of their low drug efficacy. We used the AUCs in the following analysis.

#### 2.2.2. Data Availability

The RNAseq data of 16 cell lines were downloaded from the TCGA database, and the transcriptome data of the other 31 cell lines were sequenced with the Illumina platform and processed using the same pipeline in CCLE for data compatibility. Gene expression was estimated using MMSEQ software. In the 50 cell lines, 47 had RNAseq-based gene expression data (Appendix A). The AUCs were normalized using the z-score method (mean = 0 and sd = 1); cell lines with z-score > 0.5 were defined as insensitive and cell lines with z-score < −0.5 were defined as sensitive, which corresponds to the original AUC values of 3.39 and 2.86. Finally, we obtained 13 insensitive and 15 sensitive cell lines.

Cisplatin sensitivity was analysed after 72 h of exposure with the Cell-Titer Glo assay and IC50 was calculated (data were extracted from XenoBase, Crownbioscience). We performed Pearson correlation analysis between gene expression and cisplatin sensitivity.

Biomarker discovery in cell lines and enriched pathways.

After removing genes with a high ratio of lowly expressed cell lines (>85% of cell lines with expression level <1 (expression unit is in log_2_(FPKM))), 12,694 genes were kept. Spearman’s correlation test was used to detect genes whose expression was significantly correlated with AUC. GO enrichment analysis was performed using the DAVID website https://david.ncifcrf.gov/summary.jsp (accessed on 15 November 2017). The signature genes were selected using the Boruta package in R. A linear predictor score (LPS) for each cell line of the form
(3)$$\mathrm{LPS}\left(X\right)=\underset{j}{\sum }a_{j}X_{j}$$
was calculated, where X_j_ represents the gene expression of gene j and a_j_ is the t statistics generated by the *t*-test between sensitive and insensitive cell lines. The mean and variance of the LPS distribution in sensitive and insensitive groups were estimated, and the likelihood of a cell line in each group (sensitive or insensitive) was estimated by applying Bayes’ rule so that
(4)$$P\left(X\;\mathrm{in}\;\mathrm{group}\;1\right)=\frac{\emptyset \left(\mathrm{LPS}\left(X\right);{\;\mu }_{1},{\;\sigma }_{1}^{2}\right)}{\emptyset \left(\mathrm{LPS}\left(X\right);{\;\mu }_{1},\sigma_{1}^{2}\right)+\emptyset \left(\mathrm{LPS}\left(X\right);{\;\mu }_{2},{\;\sigma }_{2}^{2}\right)}$$
where $\emptyset (x;\;\mu ,{\;\sigma }^{2}$) represents the normal density function, with mean μ and variance $\sigma^{2}$, and $\mu_{1}$, $\sigma_{1}^{2}$ and $\mu_{2},{\;\sigma }_{2}^{2}\;$ are the observed mean and variance of the LPSs within group 1 and group 2, respectively.

In addition, Gene Set Enrichment Analysis (GSEA) was performed numerically in relation to the AUC in 47 cell lines from the available unfiltered data [37]. All statistical analyses were conducted with R (version 3.1.2).

### 2.3. Gene Expression

#### 2.3.1. Transcriptome Analysis

HepG2 cells were seeded at 100,000 cells/mL and, 72 h later, treated for 4 h. Cells were then harvested, and RNA was extracted with a Direct-zol™ RNA MiniPrep kit (Zymo Research, Cat# R2051, Irvine, CA, USA) according to the manufacturer’s instructions. Transcriptome analysis was performed with Affymetrix in the Genomics Facility at SCIBLU. RMA normalized data were used to perform the analysis of enriched pathways using GSEA 4.1.0. with an interval of 10 to 500 genes per gene set and run for reactome v7.4. Gene ontology biological processes (GO-BP) were also run with filtered data, where a pre-selection was performed by only including genes differentially expressed with *p* < 0.05 and a minimum of a 1.5-fold change for downregulation in the NV651 group in comparison with the other groups. To run the enrichment analysis, a minimum of 3 genes per gene set were indicated and intermediate levels (between 3 and 8). Only pathways with a *p*-value lower than 0.05 were observed with ClueGO. Cluepedia was used for the visualization of genes. To compare gene sets that overlapped between NV651 and the control and NV651 and CsA, we set the same parameters as described above and extracted the data from the unspecified terms where the common sets are indicated.

#### 2.3.2. Quantitative PCR

HEPG2 and HUH7 cells were seeded in a 6-well plate at concentrations of 150,000 and 100,000 cells/well, respectively, in a 6-well plate and left to attach overnight. Groups were treated with either 0, 50, 100 or 500 nM of NV651 and samples were collected at 4, 8 or 24 h. At the indicated time points, trypsinized cells were collected and the total RNA was extracted using an RNeasy Mini kit (Qiagen, Cat#74106, Hilden, Germany) to the manufacturer’s instructions. The purity of the RNA was quantified using a NanoDrop 2000 spectrophotometer (Thermo Fisher Scientific, Waltham, MA, USA), and it was used for cDNA synthesis according to the manufacturer’s instructions (High-Capacity cDNA Reverse transcription kit, Cat#4368814, Applied Biosystems, Waltham, MA, USA). Quantitative PCR was performed with a QuantStudioTM 7 Flex System using SYBR^®^Green Reagent (Applied Biosystems, Waltham, MA, USA). The genes and their forward and reverse primers are indicated in Appendix A. Relative quantification was conducted following the ΔΔCt method [38] by normalizing to the housekeeping gene, GAPDH, and to the control group.

#### 2.3.3. Gene Silencing

One day after the seeding of 10,000 HEPG2 cells in a 96-well plate, the cells were treated with siRNA representing scrambled Control siRNA, CypA siRNA (Cat#sc-142741), CypB siRNA (Cat# sc-35146) and CypJ siRNA (Cat#sc-94419) at 200 nM, which were obtained from Santa Cruz Biotech, Dallas, TX, USA. On day 2, one set was dedicated for CypA, CypB and CypJ and β-actin mRNA analyses by RT-PCR. The PCR primers were obtained from Invitrogen (Appendix A). The β-actin primer was included in every PCR plate to account for sample variations. The mRNA level of each sample was normalized to that of β-actin mRNA. Cell cultures were treated with trypsin after six days, and viable detached cells were counted using a LIVE/DEAD^®^ Viability/Cytotoxicity Kit * for mammalian cells * (Invitrogen, Cat#L3224, Waltham, MA, USA) by flow cytometry according to the manufacturer’s instructions.

### 2.4. Proliferation Assay

#### 2.4.1. Acumen

HEPG2 cells were seeded in a 96-well plate at a concentration of 2000 cells/well. Groups were treated with sorafenib diluted in DMSO at concentrations of 0, 100, 200, 400, 800 or 1600 nM and NV651 at concentrations of 0, 5, 10, 15 or 20 nM. The cells were then exposed for 7 days and an acumen proliferation assay was performed as previously described [36].

#### 2.4.2. Resazurin Proliferation Assay

HEPG2 and HUH7 cells were seeded at a concentration of 1000 cells/well in a 96-well plate and left to attach overnight. The cells were then treated with NV651 at concentrations of 0.0, 0.01, 0.02, 0.05 and 0.1 µM in combination with Cisplatin at 0.0, 1.0, 2.0, 5.0 and 10 µM. Proliferation was analysed at 72 h with resazurin (Resazurin 0.01% Sigma Aldrich, Cat# R7017-1G, St. Louis, MO, USA) at a wavelength of 530/590 nm. The equivalent volume of the highest concentration of compound solvents (DMSO, Saline (0.9% NaCl in MilliQ) or MilliQ-water) was used as a control.

#### 2.4.3. Trypan Blue

HepG2 and HUH7 cells were seeded at a concentration of 15,000 cells/well in a 24-well plate 24 h before treatment and left to attach overnight. Cells were treated with 0, 0.05 and 0.1 µM of NV651 in combination with 0, 5 and 10 µM of Cisplatin and stained after 72 h of treatment. The total cell number was quantified with a Neubauer Chamber and the percentage of viable cells was estimated by the total cell number and viable cells, negative for Trypan Blue staining (Sigma Aldrich, Cat# T8154, St. Louis, MO, USA).

### 2.5. Fluorocytometry

#### 2.5.1. Quantification of Mitochondrial Membrane Potential and Cell Permeability

HEPG2 and HUH7 cells were seeded at 1.5 × 10^5^ cells/well and 1 × 10^5^ cells/well, respectively, 24 h prior to treatment. Groups were treated with 0.0, 0.05 and 0.1 µM of NV651 in combination with 0, 5 and 10 µM of Cisplatin for 72 h. The supernatant and trypsinized cells were collected for analysis. Viability was quantified using propidium iodide (PI) (Thermo Fisher Scientific, Waltham, MA, USA) at a concentration of 1 μg/mL to evaluate the plasma membrane integrity and mitochondrial membrane potential with 3,3′-dihexyloxacarbocyanine iodide (DiOC(6)3) (Molecular Probes–Invitrogen, Waltham, MA, USA) staining at a concentration of 40 nM.

#### 2.5.2. Cell Cycle Analysis

HEPG2 and HUH7 cells were seeded at 1.5 × 10^5^ cells/well and 1 × 10^5^ cells/well, respectively, 24 h prior to treatment. For the quantification of the DNA content, cells were treated with 0, 0.05 and 0.1 µM of NV651 in combination with 0, 5 and 10 µM of Cisplatin. Samples were collected at 12, 24 and 48 h, followed by fixation with 70% (*v*/*v*) ethanol. DNA was stained with 50 µg/mL PI with RNase (100 µg/mL) (Sigma-Aldrich, Cat# R4875-100MG, St. Louis, MO, USA). The SubG_1_ fraction was measured for the quantification of the apoptotic cell fraction.

### 2.6. DNA Damage: Alkaline Comet Assay

HEPG2 cells were seeded at a concentration of 100,000 cells/well and HUH7 cells at a concentration of 66,666 cells/well in a 6-well plate and left to attach overnight. The groups were then treated for 4 h with either 0 or 100 nM of NV651 followed by co-treatment with cisplatin at 0 or 10 µM for two hours. After the combination treatment, the cells were replenished with fresh media. Samples were taken at 0, 3, 6, 12 and 24 h. To increase the tails of the comets and thus increase the sensitivity of the method, the cells were exposed for 5 min to 0.1 mM H_2_O_2_, as hydrogen peroxide is known to induce random double-strand breaks and therefore facilitate the evaluation of the decrease in the olive tail moment of each comet [39,40], with some deviations. Cells were then trypsinized and resuspended at a concentration of 50,000 cells/mL. The alkaline comet assay was based on Wu and Jones [41] and Olive and Banath [42], with some modifications. Briefly, cells resuspended in PBS were mixed with 1% of low-melting-point agarose (Sigma Aldrich Cat# A4018-10G, St. Louis, MO, USA) and placed on a 1% normal-melting-point agarose (Agarose Standard Saveen Werner, Cat#A1000-500, Malmö, Sweden)-pre-coated Superfrost PLUS slides (Thermo Scientific, Cat# J1800AMNZ, Waltham, MA, USA). The cells were then left to lyse overnight at 4 °C on lysis buffer containing 2.5 M NaCl, 100 mM Na_2_EDTA, 10 mM Trisma base, 0.2 N NaOH, 10% DMSO, 0.1% sodium lauryl sarcosine and 1% TX100 (pH = 10). The next day, the samples were exposed to an alkaline condition (pH > 13) by immersing the slides in electrophoresis buffer containing 300 mM NaOH and 1 mM Na_2_EDTA. Electrophoresis was performed for 30 min at 0.8 V/cm and 300 mA at 4 °C. The samples were then neutralized with neutralization buffer (400 mM Tris-HCl pH 7.5) and the DNA was stained with PI at a concentration of 2.5 µg/mL. Comets were observed with a Zeiss Axio Vert. A1 inverted microscope, and the olive tail moment (OTM) was calculated in at least 50 comets using Tritek CometScore 2.0.0.38.

### 2.7. Statistical Analysis and Synergy

Statistical analysis was performed with GraphPad Prism 9.2.0. (San Diego, CA, USA). Data were analysed by 2-way ANOVA, followed by Dunnett’s multiple comparison test * *p* < 0.05, ** *p* < 0.01 and *** *p* < 0.001. The relative inhibition of proliferation or viability (%) was calculated in comparison with the control and used for the calculation of synergy δ-scores by SynergyFinder (version 2.0).

## 3. Results

### 3.1. NV651 Decreases Proliferation in Colorectal, Liver and Pancreatic Cancer Cell Lines; Furthermore, 18 Genes Can Effectively Predict NV651 Sensitivity

We previously confirmed the antiproliferative effect of NV651 in HCC cell lines [36]. In the present study, we extended this finding and evaluated the antiproliferative effect of NV651 in 50 cell lines comprising three types of tumours: colorectal cancer (10 cell lines), liver cancer (31 cell lines) and pancreatic cancer (9 cell lines). The antiproliferative effect of NV651 was shown to not be specific for HCC, since the proliferation of other types of cancer cells was also inhibited (Appendix A). Of the 50 cell lines, 18 did not have IC50 values due to low drug efficacy. Therefore, we used the area under curve (AUC) values in the following analysis (Appendix A). The average AUC in different tumour types differed (one-way ANOVA *p*-value = 0.024) (Appendix A). The analysis of specific cancer types showed significant differences in the AUCs (Welch’s *t*-test *p*-value = 0.007) between liver and colorectal cancer, which meant that cancer type was likely to be a factor affecting the drug response. Considering the small number of cell lines tested, all cell lines were analysed. A Spearman correlation test was used to detect genes whose expression was significantly correlated with the AUC. We identified 261 genes that were significantly correlated with the AUCs of NV651 (*p*-value < 0.001). By using the Boruta algorithm, we could identify 18 signature genes among these 261 genes. Twenty repeated 10-fold cross-validations on 28 grouped cell lines using these 18 genes showed that the prediction accuracy was 89.6%. The prediction result for these 28 cell lines is shown in Figure 1A. GSEA showed enrichment in several pathways, such as transcription-coupled nucleotide excision repair (TC-NER) or global-genome nucleotide excision repair (GG-NER), which are involved in DNA damage repair (Figure 1B).

### 3.2. NV651 Affects DNA Damage Repair and Alters the Cell Cycle

To understand the acute effect of cyclophilin inhibition with NV651 on HCC, we performed transcriptome analysis to identify dysregulated pathways after target engagement. HEPG2 cells were treated for 4 h with DMSO (as a control), CsA or NV651 at 500 nM. Then, gene set enrichment analysis was conducted for all transcribed genes among the entire RMA normalized microarray data (Figure 2A,B). When we compared CsA and NV651 treatments, we observed a downregulation in the pathways involved in DNA replication and repair, as well as in the cell cycle (including mitotic pathways) (Figure 2A). Similar pathways were also downregulated when we compared the control against NV651-treated HEPG2 cells (Figure 2B). The gene ontology–biological process (GO-BP) analysis performed on genes with a significant *p*-value (*p* < 0.05) and a downregulation with NV651 treatment (fold change < −1.5) confirmed the enrichment of the DNA damage repair and cell cycle processes (Figure 2C). Overlapping of several sets was observed when comparing NV651 with both the control and CsA treatment (Appendix A).

Next, several differentially expressed genes from sets involved in cell cycle and DNA damage repair, including factors important for the repair of interstrand crosslinks, were selected. Then, gene expression was evaluated in two different HCC cell lines with different p53 statuses: HEPG2 (WT) and HUH7 (p53 non-functional). These two commonly used HCC cell lines were selected due to a lack of integrated HBV DNA. Both HEPG2 and HUH7 showed downregulation of most of the tested genes after 4 h of treatment, confirming the gene expression data (Figure 2D–F), and even lower NV651 concentrations exerted similar effects on gene expression in a time-dependent manner (Appendix A). Downregulation in the expression of most of the genes was observed for up to 24 h in HEPG2 and HUH7, with some variability between the cell lines (Appendix A). We also evaluated the potential cyclophilin involved in the antiproliferative effect of NV651 by performing gene silencing using siRNA in HEPG2 cells. siRNAs against CypA (PPIA), CypB (PPIB) or CypJ (PPIL-3) reduced the corresponding mRNA level to <10% of the control level. Treatment with a scrambled control, PPIA, PPIB or PPIL-3 siRNA did not affect the growth of HepG2 cells (Appendix A).

### 3.3. Combination of NV651 and Cisplatin Results in a Synergistic Effect on Cell Viability in HCC Cell Lines

Since sorafenib is currently one of the few treatments available for advanced HCC, and given that previous publications have reported sorafenib’s potential disturbance in the cell cycle [43], we wanted to evaluate whether NV651, with its effect on the cell cycle, could potentially have a synergistic effect with sorafenib on cell proliferation. Evaluating the proliferation effect of sorafenib in combination with NV651 in HEPG2 cells (Figure 3A) showed that the two compounds acted synergistically (Figure 3E). Studying their potential combined effect on cell death by evaluating cell membrane integrity (late apoptotic/necrotic marker) and mitochondrial membrane potential (early apoptotic marker) resulted in no significant changes in cell death (Appendix A).

Next, we proceeded to investigate the DNA damage reagents used at an intermediate stage of HCC. A combination treatment of NV651 with cisplatin, doxorubicin and mitomycin to evaluate proliferation in HEPG2 cells (Figure 3B–D) found cisplatin to be the best compound for further study as a potential combination therapy agent (Figure 3F–H). Similar synergistic effects on proliferation were also observed in HUH7 cells (Appendix A). We confirmed a synergistic effect on cell viability after 72 h of treatment using trypan blue staining and manual counting of HEPG2 and HUH7 cells treated with 0, 0.05 and 0.1 µM of NV651 in combination with 0, 5 and 10 µM of cisplatin (Appendix A). In addition, we evaluated whether PPIA, PPIB or PPIL3 could be correlated with cisplatin sensitivity in liver tissue-derived cancer cell lines. The results showed a weak correlation between gene expression and cisplatin sensitivity, indicated as IC_50_ (Appendix A).

### 3.4. NV651 and Cisplatin Activate the Intrinsic Apoptotic Pathway in HCC Cells

To evaluate whether NV651 and cisplatin could activate the apoptotic pathway, we quantified the decrease in mitochondrial membrane potential—an early apoptotic marker —and the decrease in the cell membrane integrity—a late apoptotic marker. This evaluation by flow cytometry indicated an increase in cell death under both concentrations used in combination in HEPG2 and HUH7 cells after 72 h of treatment (Figure 4A–C). Other apoptotic markers, such as the cleaved DNA or subG_1_ population, at earlier time-points from 12 to 48 h showed the accumulation of cells in the SubG_1,_ with a significant increase from 24 h in HEPG2 cells and 48 h in HUH7 cells (Figure 4D–G).

### 3.5. NV651 and Cisplatin Cause a Decrease in the DNA Repair Capacity of HCC Cells

Owing to the downregulation of the DNA damage repair mechanisms, including homologous recombination, we wanted to evaluate whether this increased cytotoxic capacity of cisplatin in combination with NV651 could be explained by a decrease in DNA damage repair in cells. For the analysis of crosslinks, we used a modified alkaline assay, where cells were exposed to H_2_O_2_ just before harvesting (Figure 5A,B). The percentage of crosslinks at each time point was evaluated in comparison with the control. For both cell lines, we could observe a peak after 6 h of exposure to cisplatin alone (Figure 5C,D) and a decrease in the percentage of crosslinks as the cells started to repair the DNA damage, and needed to induce a double-strand break as an intermediate step before homologous recombination could take place. On the other hand, cells exposed to both NV651 and cisplatin did not present this clear decrease in the percentage of crosslinks that was observed with cisplatin alone (Figure 5C,D).

In HEPG2, NV651 and cisplatin in combination seemed to present a higher percentage of crosslinks just after exposure to cisplatin and NV651 (t = 0 h), but reached a similar peak after around 6 h (Figure 5C). NV651 alone presented an increasing trend in the percentage of crosslinks with time, with some variability between cell lines (Figure 5C,D).

**Figure 3 cancers-14-04553-f003:**
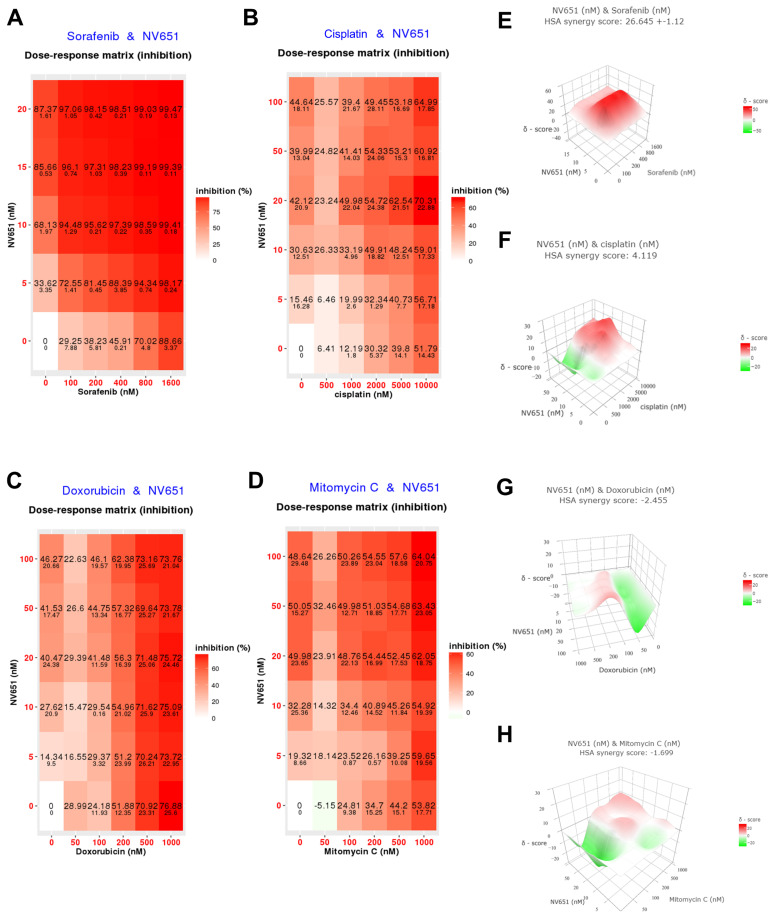
Synergistic effect of NV651 and HCC treatments in HEPG2 cells. (**A**–**D**) Effect on cell proliferation in HEPG2 cells. (**A**) HEPG2 cells were treated with NV651 in combination with sorafenib for 168 h and cell proliferation was analysed with Acumen *n* = 2 biological replicates. (**B**–**D**) Percent inhibition of NV651 in combination with Cisplatin, Doxorubicin or Mitomycin after 72 h of treatment, analysed with resazurin. *N* = 1–3 biological replicates. (**E**–**H**) Synergy score calculated with the highest single agent (HSA) from each data presented in (**A**–**D**).

**Figure 4 cancers-14-04553-f004:**
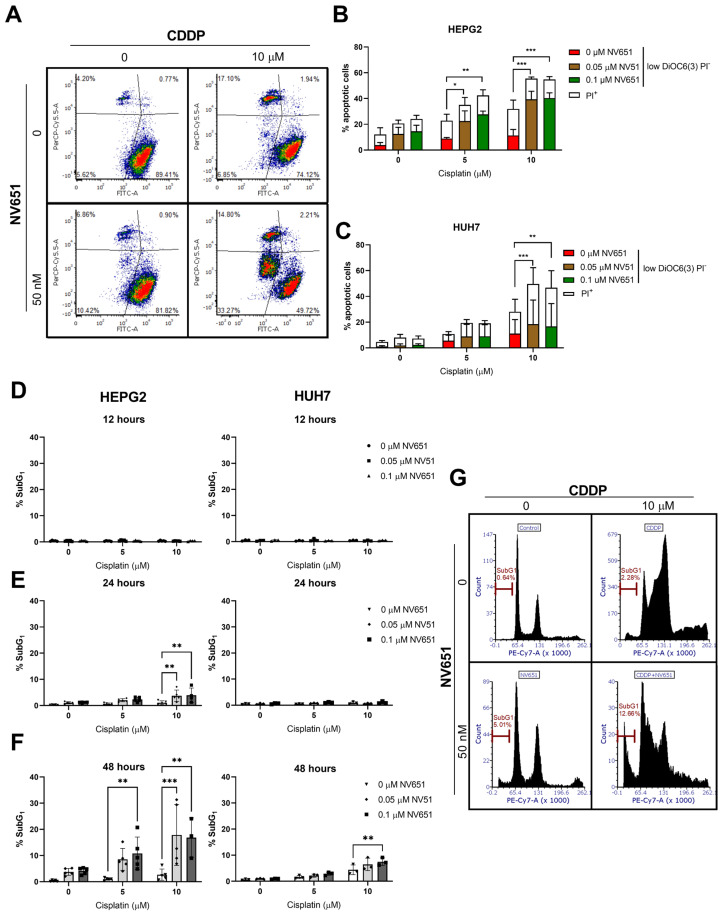
Combination effect of NV651 and Cisplatin (CDDP) on cell death. (**A**–**C**) NV651’s effect on the mitochondrial membrane potential—low DiOC6(3) levels (FITC channel) correspond to low mitochondrial membrane potential (pre-apoptotic marker)—and PI^+^cells (upper quadrants), (late apoptotic + necrotic cells) after 72 h of exposure to the combination treatment. (**A**) Representative density plot of HUH7. (**B**,**C**) Quantification of PI^+^ cells and PI^−^_low_DiOC(6)3 cells, equivalent to the low mitochondrial membrane potential in HEPG2 (**B**) and HUH7 (**C**). (**D**–**G**) SubG_1_ fraction from Figure 6 after combination treatment for 12, 24 and 48 h. (**B**,**C**) Total percentage of cell death statistically analysed by 2-way ANOVA followed by Dunnett’s multiple-comparison test. *n* = 3 biological replicates. (**D**–**F**) were statistically analysed by 2-way ANOVA followed by Dunnet’s multiple comparison test, with *n* = 3–5 biological replicates. Data are presented as the mean ± SD * *p* < 0.05, ** *p* < 0.01 and *** *p* < 0.001.

**Figure 5 cancers-14-04553-f005:**
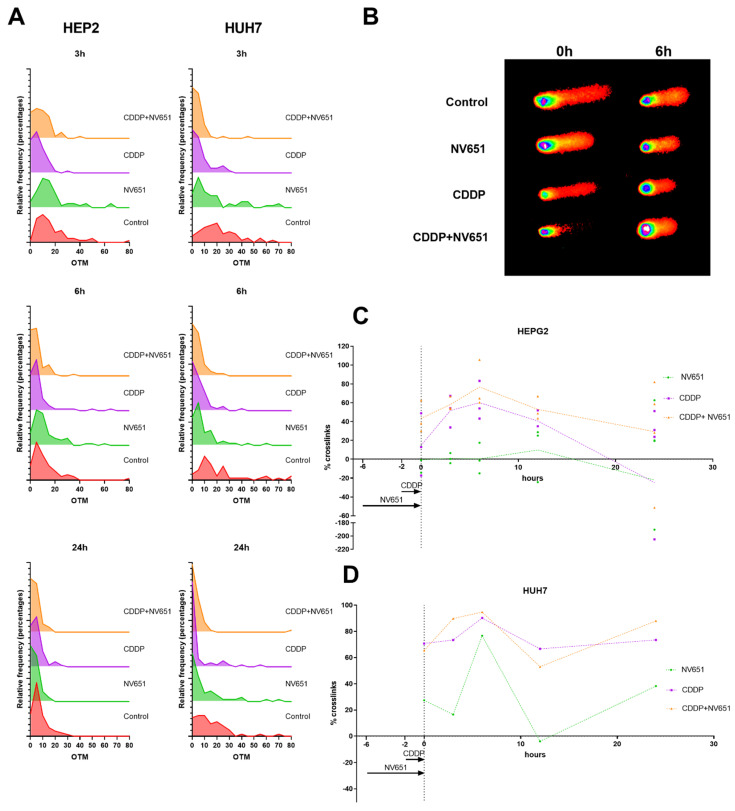
Effect of NV651 and cisplatin on DNA damage for up to 24 h in drug-free media in HEPG2 and HUH7 after 4 h of pre-treatment with NV651 followed by 2 h of cisplatin exposure. Effect on DNA damage was analysed by the alkaline COMET assay. (**A**) Frequency distribution of the olive tail moment at 3, 6 and 24 h. (**B**) Representative comets at 0 and 6 h. (**C**,**D**) Percent crosslinks calculated in comparison with the control; for each sample a minimum of 50 comets were analysed. *n* = 3–4 biological replicates in HEPG2 (**C**) and *n* = 1 biological replicate in HUH7 (**D**). Data in (**C**) are presented as the mean ± SD.

**Figure 6 cancers-14-04553-f006:**
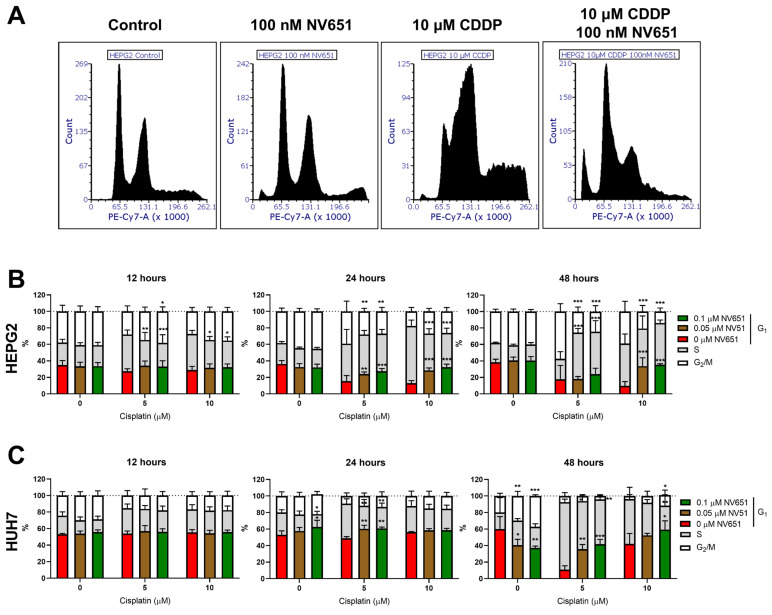
Effects of NV651 and cisplatin on the cell cycle. DNA content was analysed with PI staining in HEPG2 and HUH7 after treatment exposure for 12, 24 or 48 h. (**A**) Representative histograms of HEPG2 at 48 h of exposure; (**B**,**C**) G_1_, S and G_2_/M quantification analysed by 2-way ANOVA, followed by Dunnet’s multiple comparison test, with *n* = 3–5 biological replicates. Data are presented as the mean ± SD * *p* < 0.05, ** *p* < 0.01 and *** *p* < 0.001.

### 3.6. NV651 and Cisplatin Cause a Disturbance in the Cell Cycle

Owing to the decreased repair mechanisms for crosslinks observed in cells pre-treated with NV651 before cisplatin exposure, we wondered whether this could have an effect on the cell cycle. This effect was evaluated by analysing the DNA content of cells exposed to NV651 at 0, 50 or 100 nM and cisplatin at 0, 5 or 10 µM for up to 48 h. At 12 h, we could observe an early effect on HEPG2 cells: there was a decrease in the S phase when NV651 was present in combination with cisplatin at 5 and 10 µM concentrations (Figure 6B). At 24 h, an increase in cells in G_1_ could be observed when NV651 was combined with cisplatin in comparison with the individual cisplatin treatment. On the other hand, a decrease in the number of cells in the S phase was observed with 10 µM of cisplatin, while a decrease resulting from treatment with 5 µM of cisplatin in combination with NV651 was observed at the G_2_/M phase (Figure 6B). At 48 h, 10 µM of cisplatin in combination with NV651 accumulated cells in G_1_ and was accompanied by a decrease in the G_2_/M phase for both concentrations of cisplatin. At 5 µM of cisplatin in combination with NV651, an accumulation of cells in the S phase was observed (Figure 6A,B).

When we evaluated the effect of cisplatin in combination with NV651 in HUH7 cells (Figure 6C), another effect was observed: there were no clear changes for up to 24 h. At this time point, an increase in the G_1_ phase was observed when 5 µM of cisplatin was combined with NV651 at both concentrations used. On the other hand, a decrease in the S phase was also observed for this cisplatin concentration (Figure 6C). At 48 h, G_1_ accumulation continued under a combination of both 5 and 10 µM cisplatin with NV651; however, there was a decrease in the S phase in comparison with the individual cisplatin treatments for both cisplatin concentrations used. At the highest cisplatin concentration in combination with NV651, an increase in the G_2_/M phase was also observed in comparison with cisplatin alone (Figure 6C).

## 4. Discussion

Cyclophilins are a family of proteins overexpressed in several types of cancer [27,29,44], and their overexpression is correlated with poor prognosis [36]. This family of proteins can facilitate resistance to chemo agents, such as cisplatin, besides promoting cancer cell proliferation and metastasis [28,29,30,45]. Owing to these properties, cyclophilins have been proposed as interesting drug targets. In previous studies, cyclophilin inhibitors, such as CsA, have exhibited a decrease in proliferation in HCC [27], but the potential use of CsA as a treatment against HCC might be complicated due to its immunosuppressive activity [24]. In a previous study, we introduced NV651, a novel cyclophilin inhibitor, to treat liver cancer. NV651 was more potent than SfA and CsA in inhibiting PPIase activity and lack of immunosuppressant activity was observed in this drug class. We also confirmed its safety in normal cells. NV651 had a more potent anti-proliferative effect in HCC cell lines than sorafenib. NV651 also had the capacity to decrease tumour growth in vivo. Cell cycle perturbations accompanied by an accumulation of cells in the mitotic phase were observed upon treatment [36].

In the current study, we evaluated the effect of NV651 on several cancer cell lines. NV651 exhibited a broad anti-proliferative effect against several types of cancer cells. We found that a combination of 18 biomarkers could be used for the prediction of sensitivity with genes, such as BORA, an activator of the protein kinase Aurora A that is involved in the mitotic phase. GSEA resulted in the enrichment of several pathways, such as nucleotide excision repair (NER) or ATM, and had a negative normalized enrichment score, indicating sensitivity to NV651.

To confirm this, transcriptome analysis was performed on HEPG2 cells treated with either DMSO as a control, CsA or NV651. We observed the downregulation of several pathways involved in DNA damage repair mechanisms, such as homologous recombination, and the cell cycle, including DNA replication and mitosis. The downregulation of DNA damage repair has been previously observed with CsA treatment [34,46]. On the other hand, when we evaluated differences between the CsA treated group and NV651, we could also observe a further downregulation in the pathways involved in mitosis and DNA damage repair in NV651-treated cells. This could explain the previously observed exclusivity of the mitotic block of NV651 in HCC cell lines that was not reported with cyclosporin treatment in HCC.

Cyclophilins have been shown to potentiate therapy resistance through different mechanisms, such as the reduction of intracellular drug accumulation through increased expression of ABC transporters [31]. Therefore, we continued our study with the evaluation of potential combination treatments of NV651 with other treatments clinically used against HCC. Initially, a combination of sorafenib—one of the few treatments approved for advanced HCC—and NV651 was evaluated. Although a potential synergistic effect on cell proliferation was observed, no clear increased cell death resulted from this combination in our experiments.

Cisplatin is a well-studied chemo agent that can be used during the intermediate stage in HCC patients. Different types of DNA damage occur from the interaction between cisplatin and DNA, including inter-and intrastrand crosslinks. Cell fate will depend on the level of DNA damage produced and the capacity of the cell to repair DNA adducts [8]. Therefore, when cells are not able to repair their DNA damage effectively, cell death will often occur through the apoptotic pathway [47]. Unfortunately, intrinsic and acquired resistance to cisplatin is a common event that can be induced through different pathways, such as DNA repair or increased antioxidant capacity [48,49].

One of the main effects that was believed to increase the cytotoxicity of cisplatin treatment was due to the stalling of replication forks. During the S-phase, interstrand crosslinks can be repaired through the Fanconi anaemia pathway followed by a homologous recombination, where BRCA1/2 are important players [50,51]. Due to the importance of interstrand crosslink repairs during the S phase, one of the mechanisms of the cells to acquire cisplatin resistance is increased homologous recombination or the upregulation of genes involved in Fanconi anaemia [52,53]. Therefore, the downregulation of genes such as BRCA1 or FANCD2 (BRCA2) has been shown to increase sensitivity to cisplatin-resistant cells. Owing to these previous results, targeting the homologous recombination or Fanconi anaemia could be a strategy to overcome cisplatin resistance [53]. Under NV651 treatment, gene expression and transcriptome analysis have demonstrated downregulation of genes involved in Fanconi anaemia, as well as the homologous recombination pathway.

We hypothesized that chemo agents, such as cisplatin, doxorubicin or mitomycin, which are used during the intermediate stage, could have a synergistic effect with NV651. A clear synergistic effect could be observed in combination with cisplatin treatment. Therefore, this compound was selected for further studies. Furthermore, it was confirmed that a decrease in the mitochondrial membrane potential, an increase in the percentage of the population with cleaved DNA or subG_1_ and a decrease in the cell membrane integrity point to a decrease in cell viability due to the activation of the apoptotic pathway.

Due to the decreased DNA damage repair and the synergistic effect on cell viability with the activation of the apoptotic pathway, we analysed whether this mechanism could affect the capacity of the cells to repair crosslinks. This was investigated by the pre-treatment of cells with NV651 followed by cisplatin exposure to induce interstrand crosslinks. After the cells were left to repair, several time points were evaluated in comparison with the individual treatment with cisplatin. We observed a decrease in the repair of interstrand crosslinks when cells were pre-exposed to NV651 in comparison with the individual treatment with cisplatin.

Owing to the decreased repair of crosslinks and the importance of their repair to proper DNA replication, we then proceeded to evaluate the effect of NV651 on the cell cycle. A clear G_1_ block was observed in both cell lines. Differences in the S and G_2_/M percentages were observed between HEPG2 and HUH7, as well as in the concentration of cisplatin used. Although cisplatin cytotoxicity was believed to be mainly due to the stalling of replication forks, other pathways that play an important role in the repair of interstrand crosslinks and the resistance of the cells to this compound have been discovered. During the G_1_ phase, two different nucleotide excision repair pathways can be activated for the repair of interstrand crosslinks: TC-NER and GG-NER [54]. TC-NER is important for the repair of regions that need to be transcribed from the genome; however, the non-transcribed regions are repaired by GG-NER [55]. It has previously been discovered that TC-NER is a key pathway in the resistance to cisplatin [54], where alterations in TC-NER and translesion synthesis (TLS) can help cisplatin resistance or increase sensitivity [54,56,57]. This type of DNA damage repair in G_1_ is independent of homologous recombination or of the Fanconi anaemia pathway, but decreases the number of crosslinks that need to be repaired through the S phase, thus showing an additive effect when both mechanisms are downregulated [57]. In in vitro human ovarian carcinoma, CsA treatment decreases the expression of ERCC1, a key factor in DNA repair through NER that is linked to cisplatin resistance [58]. In addition, CsA, in combination with cisplatin, was able to ameliorate the increased expression of ERCC1 observed upon treatment with cisplatin and increase cytotoxicity [59]. In our study, NV651 downregulated different types of mechanisms used by cells to remove interstrand crosslinks. This could potentiate the effect of cisplatin and impede the appearance of cisplatin resistance. Treatment with cisplatin in cells with a functional NER will cause a block in the S phase and proceed to a G_2_/M block after 48 h of cisplatin treatment. On the other hand, a defective NER function would induce a blockage of cells in G_1_ and an accumulation of cells in the subG_1_ phase due to increased apoptosis [60]. Therefore, these results could explain the G_1_ block observed with NV651 in combination with cisplatin in both HCC cell lines, as well as S and G_2_/M accumulation under individual cisplatin treatment. Taken together, we believe that the synergistic effect of cisplatin and NV651 can be caused by DNA damage induced by cisplatin and the lack of a proper DNA repair system due to NV651 exposure. The activation of the DNA damage response might lead to cell cycle arrest as the cell tries to repair the DNA; however, due to the inability of cells to successfully do so, intrinsic apoptosis will be activated. Further experiments would need to take place to better understand the link between DNA damage repair and cell cycle arrest.

The discovery and confirmation of which cyclophilin is involved in the mechanism of action of NV651 by causing gene silencing with siRNA is also of great interest. However, some aspects of cyclophilins and NV651 need to be taken into consideration when evaluating the effect of silencing a single cyclophilin or when evaluating the potential correlation between its expression and cisplatin sensitivity. Multiple cyclophilins have been identified in the human genome. These cyclophilins share a common domain known as the cyclophilin-like domain or CLD of approximately 109 amino acids [19]. The silencing of cyclophilin A, B and J using siRNA resulted in no clear effect on the total cell number. We believe that these results could be explained by the overlapping functions of these proteins that lead to other cyclophilins compensating when one of the cyclophilins is silenced. Considering this and the fact that NV651 is able to inhibit the PPIase activity of all cyclophilins tested, we believe that NV651’s mechanism of action is not dependent on the inhibition of one single cyclophilin, but on the inhibition of several cyclophilins. Further studies would need to be performed to confirm this hypothesis.

In this study, we evaluated the effect of NV651 in combination with sorafenib, mitomycin, doxorubicin and cisplatin. Although we expected a synergy in all chemo agents used, no clear synergy was observed with mitomycin or doxorubicin. Future experiments should include a detailed evaluation of the potential pathways involved in DNA damage repair and the type of DNA damage induced that could explain these differences in synergy between chemo agents and NV651. It would also be of great interest to evaluate whether NV651 could present synergistic anticancer properties with other platinum-containing cisplatin analogues, such as carboplatin.

In recent years, ICIs have emerged as interesting new targets [15]. Studies showing alterations in the DNA damage repair pathways as potential biomarkers for sensitivity to ICIs have been conducted [61]. It has been hypothesized that ICIs combined with TACE could be a potentially effective combination treatment due to the release of tumour-associated antigens that increase the immune response, which could lead to a synergistic effect [62]. Currently, there are several clinical trials evaluating the monoclonal antibody nivolumab, which targets the programmed cell death-1 (PD-1) receptor in patients with intermediate HCC (NCT04268888). Specifically, cisplatin, together with gemcitabine, will also be evaluated in combination with bevazicumab and atezolizumab in stage III–IV unresectable HCC patients (NCT05211323). We previously evaluated the effect of one of our cyclophilin inhibitors, NV556, in two liver fibrosis animal models and reported no clear effect on inflammatory cell infiltration when a chronic injury takes place [63]. Although the experimental model would need to be prolonged for the appearance of tumours, the lack of an anti-inflammatory response in this model could indicate that at least no antagonist effect with ICIs would take place. On the other hand, the potential synergistic effect of NV651 with cisplatin could lead to an indirect increase in the immune response. Further validation is needed.

## 5. Conclusions

The novel combination treatment of NV651 and cisplatin showed a potential synergistic effect on cell viability and increased activation of apoptosis. This could provide a solution to the dose limitation of chemotherapies, enabling potent treatment effects while minimizing side effects and the risk of developing chemoresistance.

## 6. Patents

Gronberg et al. (2020). Use of sanglifehrin macrocyclic analogues as anticancer compounds (US 10,857,150 B2). United States Patent.

## Figures and Tables

**Figure 1 cancers-14-04553-f001:**
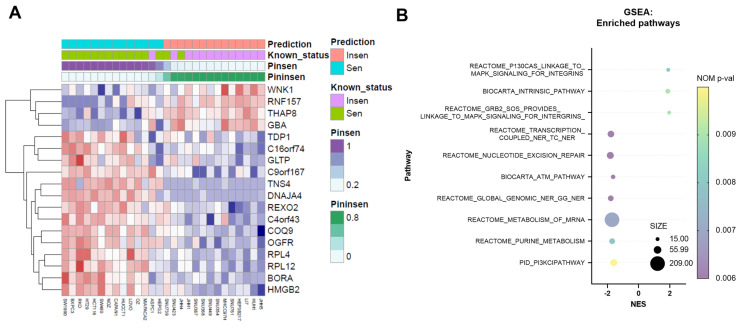
Biomarker discovery for NV651. (**A**) Drug efficacy prediction results using 18 signature genes in 28 grouped cell lines. The probability of being in the sensitive group is indicated as Pinsen and probability of being in the insensitive group is indicated as Pininsen. The drug response of 25 cell lines was correctly predicted. (**B**) Gene set enrichment analysis (GSEA) in 47 cell lines indicating the NES (normalized enrichment score), size and nominal *p*-value. The correlated genes were enriched in 10 pathways, with a NOM *p*-value of <0.01.

**Figure 2 cancers-14-04553-f002:**
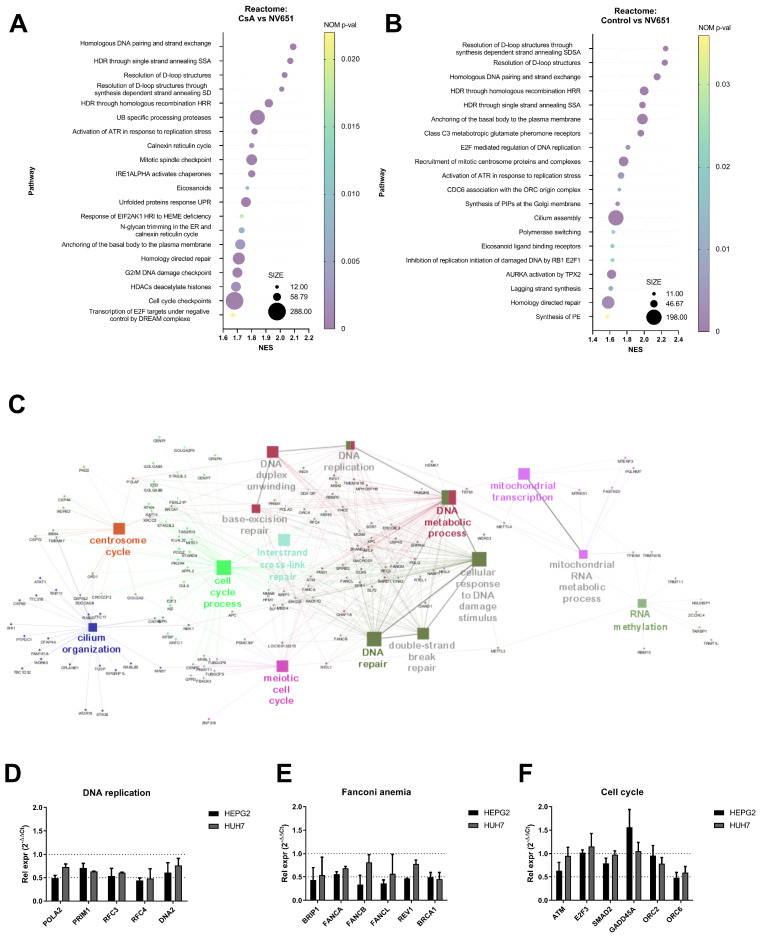
NV651 effect on gene expression. (**A**–**C**) Transcriptome was analysed after 4 h of 500 nM NV651, CsA or control treatment in HEPG2 cells. (**A**,**B**) Top 20 genes sets for GSEA in the reactome for CsA versus NV651 (**A**) and control versus NV651 (**B**). (**C**) Cytoscape with the GO biological process at an intermediate level (3–8) with the significant genes indicated (kappa score ≥ 0.4). (**D**–**F**) NV651’s effect on gene expression after 4 h of exposure in HEPG2 and HUH7 cells. mRNA levels of the indicated genes were analysed by qPCR. Data in (**D**,**F**) are presented as the mean ± SD of the relative expression in *n* = 2 biological replicates.

## Data Availability

The RNAseq data of 16 cell lines were downloaded from the TCGA database for biomarker analysis. Cisplatin sensitivity was retrieved from XenoBase, Crownbioscience https://xenobase.crownbio.com/ (accessed on 13 June 2022). Enriched pathways were evaluated from our transcriptome data using GSEA 4.1.0.

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
