# Peer review of "Synergistic Effects of Sanglifehrin-Based Cyclophilin Inhibitor NV651 with Cisplatin in Hepatocellular Carcinoma"

_cancers, 2022, doi:10.3390/cancers14194553_

Round 1

Reviewer 1 Report (Previous Reviewer 2)

Thank you for your email.
The authors addressed all the queries and issues we raised.
We recommend Acceptance.

Author Response

Response: We thank the Reviewer for the careful review of our manuscript.

Reviewer 2 Report (New Reviewer)

Kindly follow the instruction by the editor.

Author Response

Response: We thank the Reviewer for the careful review of our manuscript.

Reviewer 3 Report (New Reviewer)

Among all malignancies, hepatocellular carcinoma (HCC) is one of the deadliest. Despite significant advancements in detection and treatment, the incidence of HCC is increasing. Trans-arterial chemoembolization (TACE) and oral dosage with the kinase inhibitor sorafenib are two possible treatments for more advanced stages. However, drug resistance, drug inefficiency, and chemotoxicity become evident. Because of this, neither existing ablation therapies nor chemotherapy significantly improve the course of this deadly disease. Thus, further study is required to develop effective therapies for treating liver cancer. In this manuscript, the authors demonstrated the synergistic effect of NV651 with cisplatin in HCC. Discussion is well written. However, I have some concerns from this article.

 Why did the authors use 24h time point for qPCR while NV651 examined for 72h in most experiment?

The authors looked at the Acumen assay for sorafenib and NV651 treatment in HEPG2 cells for the proliferation, whereas the Resazurin assay employed Cisplatin and NV651 for HEPG2 and HUH7 cells. Why there is a discrepancy between the experiment and also the drug doses of NV621 varied in both tests.

 Trypan blue was employed for % viability, which is acceptable, but it's unclear how they determined the two set dosages of each drug for 72 hours. If the trypan blue experiment is the basis for the synergistic effect, then what was the combination index (CI) value? Cell viability should be tested using the MTT assay at various time points and dosages. Because the article title shows that NV651 and cisplatin work together synergistically. 

 Figure 2D: p-value should be calculated.

 Figure D-F: NV651 effects on gene expression demonstrated but what the results were after 8 and 24 hours of exposure, as they stated in the methodology.

 Figure 4A: The legends should explain what the quadrants in the figure represent.

Results should include the percentage of cell cycle and apoptosis.

 Although the authors displayed a cell cycle assay, a western blot of the cell cycle checkpoint will provide a better description of this work.

 The authors should report the experiment using a mechanistic pathway approach as they are not demonstrating the effect in an in vivo model. For a better understanding of the combination effect in HCC, a model should also be provided.

Author Response

Response: We thank the Reviewer for the careful review of our manuscript and the valuable and constructive suggestions.

 Why did the authors use 24h time point for qPCR while NV651 examined for 72h in most experiment?

Response: Our aim performing transcriptome analysis at four hours was to evaluate the direct changes of inhibiting the PPIase activity of the cyclophilins. We have included this information in the results section. Downregulation of several genes involved in DNA damage repair and cell cycle were confirmed with the same concentration and time point used for the transcriptome analysis. This analysis was complemented by a time-response effect up to 24 hours with lower concentrations: 50 and 100 nM (Figure S2), the same concentrations that were used to evaluate synergy with cisplatin in relation to cell viability. To analyze cell viability, we decided to use the 72 hours timepoint, as cell death takes longer to be observed, which was demonstrated with the increase of SubG1 through time (Figure 4).

The authors looked at the Acumen assay for sorafenib and NV651 treatment in HEPG2 cells for the proliferation, whereas the Resazurin assay employed Cisplatin and NV651 for HEPG2 and HUH7 cells. Why there is a discrepancy between the experiment and also the drug doses of NV621 varied in both tests.

Response: The evaluation of the potential synergistic effect of Sorafenib and NV651 was performed in Shanghai chempartner by direct cell counting with Acumen, a method performed in this company. The concentrations used of NV651 and sorafenib were optimized for that specific combination of compounds. Our main aim was to increase cell death when both compounds were combined, therefore after the lack of synergy on cell death, we decided to try other potential combinations such as cisplatin, mitomycin and doxorubicin. One of the methods available and commonly used in our lab was Alamar blue (resazurin) which acts as a redox indicator that in the reduced form is highly fluorescent. Metabolically active cells will transform resazurin into its reduced form and the proportional intensity of fluorescence can be measured to quantify the number of living cells. In our pilot experiment up to 100 µM for cisplatin and lower concentrations for NV651 were tested. Based on this data, we selected the range of concentrations that showed synergy for both compounds in this methodology and could also increase apoptosis.

Trypan blue was employed for % viability, which is acceptable, but it's unclear how they determined the two set dosages of each drug for 72 hours. If the trypan blue experiment is the basis for the synergistic effect, then what was the combination index (CI) value?

Response: We evaluated the concentrations needed to decrease cell viability, specifically when cisplatin was combined with NV651. These concentrations were selected based on the results from Figure 3 and Figure S5.  A pilot experiment was then performed with a detailed titration of NV651 co-treated with cisplatin and analyzed by mitochondrial membrane potential and PI+ cells to evaluate the effect on cell death and select the most interesting concentrations for cisplatin and NV651. Figure S6 was used as a confirmation of the decreased cell viability with the same selected concentrations (5 and 10 µM of cisplatin and 50 and 100 nM of NV651).

Since there is no standardised model for determining synergy between compounds, we confirmed the synergistic activity of cisplatin and NV651 with synergyfinder using two different models (Figure S6). The first reference model used in this study to quantify synergy by δ-score was highest single agent (HSA) which determines synergy by calculation of excess over the maximum single drug response. To confirm the results, a second model was used: Bliss, which calculates the expected multiplicative effect of each single drug as if they acted independently. We have already calculated combination index (CI) but based on the presented data above, we believe that CI does not add additional information and it is less informative.

Cell viability should be tested using the MTT assay at various time points and dosages. Because the article title shows that NV651 and cisplatin work together synergistically. 

Response: We did not perform MTT assay as it presents a similar principle as Alamar blue, by detecting metabolically active cells to analyse cell viability. We hope that the reviewer agrees that the effect of NV651 combined with cisplatin in HEPG2 and HUH7 cells presented in Figure 3 and Figure S5 is enough to suggest synergy between these two compounds presented.

 Figure 2D: p-value should be calculated.

Response: The main aim of figure 2D was to confirm the results by qPCR from the transcriptome analysis by using the same NV651 concentration and time point. We performed this experiment twice as a confirmation.

 Figure D-F: NV651 effects on gene expression demonstrated but what the results were after 8 and 24 hours of exposure, as they stated in the methodology.

Response: The aim to analyse up to 24 hours was to evaluate whether the effects observed at 4 hours would be maintained. We briefly mention the results of this supplementary figure in the results section.

 Figure 4A: The legends should explain what the quadrants in the figure represent.

Response: We have now added this information in the legend.

Results should include the percentage of cell cycle and apoptosis.

Response: The effect on cell cycle and apoptosis is presented as percentage of the total number of analysed cells as can be seen in figures 4 and 6.

 Although the authors displayed a cell cycle assay, a western blot of the cell cycle checkpoint will provide a better description of this work.

Response: The authors are aware of the importance of cyclins and cyclin-dependent kinase among other markers. We have previously analysed markers such as cyclin B1 and phosphorylation of histone 3 with western blot for individual treatment of NV651 but the results have not been successful as HEPG2 cells present two different populations: diploid and tetraploid. This can be observed in figure 4G and it becomes more evident with NV651 treatment as it increases the G2/M phases of diploid and tetraploid populations. We have mentioned this limitation in the manuscript.

The authors should report the experiment using a mechanistic pathway approach as they are not demonstrating the effect in an in vivo model. For a better understanding of the combination effect in HCC, a model should also be provided.

Response: Due to the complexity of both compounds, we have summarized our hypothesis explaining why cisplatin and NV651 lead to a synergistic effect on cell death.

Round 2

Reviewer 3 Report (New Reviewer)

All questions have been addressed by the authors.

This manuscript is a resubmission of an earlier submission. The following is a list of the peer review reports and author responses from that submission.

Round 1

Reviewer 1 Report

Serrano et al demonstrated a wide range of genomic data analysis in a combination with cell-based assays to determine a synergistic effect of cyclophilin with a conventional cancer chemotherapy cisplatin. Considering the clinical significance of hepatocellular carcinoma (HCC), this paper seems to be important. However, because of many flaws in each experimental design and the ways of data presentation, this reviewer often encountered technical or scientific difficulty to follow the authors’ conclusions. 

Major concern

The etiology of chronic HCC is various, but mainly due to alcohol and the hepatitis C virus. The HCC cell lines that the authors used in this paper should be first categorized based on these criteria and find out any correlation between cyclophilin gene/protein expression and cisplatin sensitivity. As the positive control for NV651, the authors should use siRNA against cyclophilin. If this is the case, the authors need to show what cyclophilin (-A or -D) should be inactivated. If the authors clarify this as well in different HCC cell lines in the presence and absence of cisplatin, the potential values of the paper will be increased.

Minor issues

Line 19: We demonstrated that the novel cyclophilin inhibitor, NV651, affected several pathways related to the cancer cell cycle and DNA repair mechanism.

Reviewer 1’s comments: How did the authors rule out the possibility of the off-target effect by NV651? How did they determine the minimum dose of this drug? If cyclophilin is a peptidase, it must have its own Km value using a typical peptide as a substrate. The authors should also show us at least a dose-dependent titration experiment.

Line 20: …, NV651, affected several pathways…

Reviewer 1’s comments: ‘affected’ automatically contains both ‘positive’ and ‘negative’ effects. Clarify either. If both, say so.

Line 22: …. due to its significant increase in cancer cell death compared to cisplatin alone.

Reviewer 1’s comments: What mode of cancer-cell death? Cisplatin-sensitive or resistant? If the authors argue for a ‘synthetic lethal effect,’ they are committed to identifying the 2nd pathway leading to cisplatin sensitivity. The authors should show whether the synthetic lethal effect is specific to cisplatin. How about carboplatin?

Line 28: …..Our aim in this study was to evaluate changes…

Reviewer 1’s comments: Remove "in this study" as it is obvious. "Here, we aim to evaluate…" sounds more straightforward.

Line 29: ….in order to explore suitable combination treatment approaches.

Reviewer 1’s comments: Combination with ‘what’? NV651 with several anticancer chemotherapeutics, including cisplatin? If so, say so.

Line 30-31: Our results showed that the novel cyclophilin inhibitor NV651 affected several pathways related cancer cell cycle and DNA repair.

Reviewer 1’s comments: This is essentially the same sentence as the one in lines 19-20. Rephrase it to avoid redundancy. Remove it.

Line 32-33: NV651 showed a synergistic effect on cell proliferation with a significant increase in cell death in combination with the DNA cross-linking agent cisplatin.

Reviewer 1’s comments: This sentence is confusing. NV651 synergistically increases cell proliferation and cell death in the presence of cisplatin???

Line 34: … This combination also affected cell cycle progression and …

Reviewer 1’s comments: Again, ‘affected’ automatically contains both ‘positive’ and ‘negative’ effects. Clarify either. If both, say so.

Line 283-285: …. The antiproliferative effect of NV651 was shown to not be specific for HCC since the proliferation of other types of cancer cells was also inhibited (Table S1).

Reviewer 1’s comments: There is no table legend for Table S1. Moreover, even within the liver cancer cell lines, they are not well ordered. This reviewer also noticed that each value shows a wide range of variety. How could the authors conclude an ‘antiproliferative’ effect based on this table? Show us an average value and its SD value within the cancer cell lines derived from the same organ. These cell lines seem to be listed based on different ‘Groups’ (including Insen, N/A, and Sen). What do they mean? If insensitive and sensitive, what criteria (i.e., cut-off value to inhibit cell growth, etc.) did the authors employ to determine the sensitivity vs. insensitivity? There are too many cell lines (22 out of 50), the sensitivity of which remains N/A. The title of Table S1 does not seem to reflect the content as there is no genomic data, either. Overall, this reviewer judged that Table S1 is not informative, although it may serve as useful lab notebook data.

Line 287: … Significant genes were enriched in 34 GO terms and…

Reviewer 1’s comments: What does ‘GO terms’ mean? This reviewer used a ‘Find’ command in the MS Word program, but could not find this in any other place but here in this paper.

Fig. 1:   Fonts in each figure panel are too small to read. Fig. 1’s legend is not very informative. Pinsen vs. Pininsen? What are they? The scope of the journal “Cancers” is pretty wide, so many of the readers of this journal including myself are not very familiar with these technical terms. A total of 18 signature genes in 28 cell lines? How would you evaluate they are ‘signature’ genes? Clarify your own criteria, then we can follow your logic.  

Fig. 2:   Surprisingly, in panels D-F, there is no vehicle control. If you do not have it, how could we interpret the result?

Fig. 3:   How do the authors distinguish the difference between the ‘synergistic’ effect and the ‘additive’ effect? Again, the fonts in this figure are also too small to follow.

Fig. 4-6: The authors conducted and collected a pretty interesting set of functional data here. However, is there any rationale to use HepG2 and Huh7 as the representative two liver cancer cell lines? Again, the fonts in this figure are also too small to follow.

Reviewer 2 Report

A very interesting study exploring an under-discussed and frequently overlooked topic in hepatocellular carcinoma. Certainly, the study assesses an interesting topic, but some points need to be addressed. 

The manuscript is quite well written and organized. English could be improved.

Figures and tables are comprehensive and clear.

The introduction explains in a clear and coherent manner the background of this study.

We suggest the following modifications:

  • Introduction section: although the authors correctly included important papers in this setting, we believe some recent studies discussing emerging and novel treatment options in HCC should be cited within the introduction ( PMID: 34431725 ; PMID: 34429006), only for a matter of consistency. We think it might be useful to introduce the topic of this interesting study.
  • Methods and Statistical Analysis: nothing to add.
  • Discussion section: the authors should better discuss this part. In fact, since the current landscape of HCC is seen impressive revolution, the potential role of cisplatin needs to be better highlighted.

However, we think the authors should be acknowledged for their work. In fact, they correctly addressed an important topic in HCC, the methods sound good and their discussion is well balanced.

One additional little flaw: the authors could better explain the limitations of their work, in the last part of the Discussion.

We believe this article is suitable for publication in the journal although some points need to be addressed. The main strengths of this paper are that it addresses an interesting and very timely question and provides a clear answer, with some limitations.

We suggest a linguistic revision and the addition of some references for a matter of consistency. Moreover, the authors should better clarify some points.
